# Infectious Norovirus Is Chronically Shed by Immunocompromised Pediatric Hosts

**DOI:** 10.3390/v12060619

**Published:** 2020-06-05

**Authors:** Amy Davis, Valerie Cortez, Marco Grodzki, Ronald Dallas, Jose Ferrolino, Pamela Freiden, Gabriela Maron, Hana Hakim, Randall T. Hayden, Li Tang, Adam Huys, Abimbola O. Kolawole, Christiane E. Wobus, Melissa K. Jones, Stephanie M. Karst, Stacey Schultz-Cherry

**Affiliations:** 1Department of Infectious Diseases, St. Jude Children’s Research Hospital, Memphis, TN 38105, USA; amy.davis2@stjude.org (A.D.); valerie.cortez@stjude.org (V.C.); ronald.dallas@stjude.org (R.D.); jose.ferrolino@stjude.org (J.F.); pamela.freiden@stjude.org (P.F.); gabriela.maron@stjude.org (G.M.); hana.hakim@stjude.org (H.H.); 2Department of Molecular Genetics and Microbiology, University of Florida, Gainesville, FL 32611, USA; marco.grodzki@ufl.edu; 3Department of Pathology, St. Jude Children’s Research Hospital, Memphis, TN 38105, USA; randall.hayden@stjude.org; 4Department of Biostatistics, St. Jude Children’s Research Hospital, Memphis, TN 38105, USA; li.tang@stjude.org; 5Department of Microbiology and Cell Science, University of Florida, Gainesville, FL 32611, USA; a.huys@usask.ca (A.H.); mmk@ufl.edu (M.K.J.); 6Department of Microbiology and Immunology, University of Michigan, Ann Arbor, MI 48109, USA; akolaw@med.umich.edu (A.O.K.); cwobus@umich.edu (C.E.W.)

**Keywords:** norovirus, immunocompromised host, infectious virus shedding, genotype, diarrhea, asymptomatic

## Abstract

Noroviruses are a leading cause of gastroenteritis worldwide. Although infections in healthy individuals are self-resolving, immunocompromised individuals are at risk for chronic disease and severe complications. Chronic norovirus infections in immunocompromised hosts are often characterized by long-term virus shedding, but it is unclear whether this shed virus remains infectious. We investigated the prevalence, genetic heterogeneity, and temporal aspects of norovirus infections in 1140 patients treated during a 6-year period at a pediatric research hospital. Additionally, we identified 20 patients with chronic infections lasting 37 to >418 days. Using a new human norovirus in vitro assay, we confirmed the continuous shedding of infectious virus for the first time. Shedding lasted longer in male patients and those with diarrheal symptoms. Prolonged shedding of infectious norovirus in immunocompromised hosts can potentially increase the likelihood of transmission, highlighting the importance of isolation precautions to prevent nosocomial infections.

## 1. Introduction

Noroviruses are highly infectious gastrointestinal viruses that are transmitted via multiple routes, including ingestion of food or water with fecal contamination and inhalation of infectious aerosols produced during episodes of emesis [1,2]. Noroviruses are highly resistant to inactivation by freezing, heating, or detergent-based cleaning fluids [3]. It is estimated that 1 out of every 5 cases of gastroenteritis is attributed to a norovirus [4,5], making these viruses a leading cause of childhood diarrhea and foodborne outbreaks of disease worldwide [4,5,6]. Noroviruses are small, nonenveloped viruses belonging to the family *Caliciviridae*. They have a single-stranded RNA genome and are segregated into 10 genogroups, designated GI to GX, with GI-, GII-, and GIV-containing strains associated with human gastroenteritis [3,7,8]. These genogroups are further subdivided into genotypes (e.g., a genogroup I, genotype 1 virus would be referred to as GI.1). Until recently, norovirus classification was based on the identity of its genogroup and genotype. Today, norovirus nomenclature can additionally represent nucleotide sequence variation within the RNA-dependent RNA polymerase region of the genome, where strains can be grouped into polymerase (P)-groups and P-types. Most norovirus outbreaks since 1996 have been associated with a single genotype, GII.4, with at least six norovirus pandemics having been reported during this period [7,9]. With some exceptions [10,11], a new GII.4 strain has emerged every 2–4 years [9,12], with the last new pandemic strain, GII.4 Sydney, emerging in 2012. Noroviruses frequently recombine, and there is a recombination hotspot between *open reading frame* (*ORF*)*1* and *ORF2* [13]. The original GII.4 Sydney strain had a GII.Pe *ORF1* gene (referred to as GII.Pe-GII.4 Sydney), whereas a new recombinant strain, GII.P16-GII.4 Sydney, became prevalent in 2015, and it remains the predominant strain in the United States [14,15]. Strain GII.P16-GII.2 has also been commonly found in circulation since 2016 [14]. The reservoir from which new norovirus strains emerge is unknown, but it has been speculated that immunocompromised hosts could be a source of emergent strains because of their higher risk of chronic infection [9].

Diarrheal symptoms accompanied by norovirus infections in healthy individuals resolve themselves in a few days [16], but immunocompromised patients can develop chronic symptomatic norovirus infections lasting weeks or even years [7], and they shed viral genomes for much longer [17,18]. The lack of cell culture models for these viruses has historically precluded testing of whether the shed virus is infectious. This is a key question for public health in terms of evaluating the significance of prolonged shedding for inter-host transmission dynamics [19] and, specifically, whether the duration of isolation precautions after norovirus infection should be extended for immunocompromised individuals [20]. Herein, we describe the prevalence, genetic heterogeneity, and temporal aspects of norovirus infections at a pediatric research hospital. In the course of this study, we identified a cohort of chronically infected patients. Based on our results obtained using a novel human norovirus propagation system, we report that both asymptomatic and symptomatic chronic infections are associated with the shedding of infectious virus.

## 2. Materials and Methods

### 2.1. Study Population

This study used both remnant and prospectively collected stool samples from a total of 1140 pediatric patients treated at St. Jude Children’s Research Hospital in Memphis, Tennessee, USA, between January 2012 and April 2018. Stool samples were collected under 3 different Institutional Review Board (IRB)-approved protocols: ALLGUT (2012–2016), EPIGUT (2015–2018), and FAECIS (2012–2016). The ALLGUT study has been previously described [21]. Briefly, patients with acute lymphoblastic leukemia were enrolled prior to chemotherapy, and 1 to 4 samples were collected per patient at set time points during their chemotherapy, which may or may not have co-occurred with symptomatic gastroenteritis. A total of 513 stool samples from this protocol were included in this analysis. The EPIGUT study enrolled patients with hematologic malignancies, solid tumors, and undergoing allogeneic hematopoietic cell transplants (HCT), who contributed 1349 stool samples collected at protocol-defined time points over the course of one year, regardless of symptoms. Additional samples were collected every 0, 7, and 14 days during diarrheal episodes. The FAECIS retrospective cohort included remnant stool samples, originally submitted for clinical diagnostic testing of patients with diarrheal symptoms. A total of 2782 samples were collected and analyzed from the FAECIS study. All 3 cohorts occurred during a recent but overlapping period of time, which provided not only a similar epidemiological period for our observations, but also similar demographic characteristics in our population of immunocompromised patients. The samples available from the 3 protocols were combined for this study in order to (1) increase our sample size and observation time per patient, (2) include asymptomatic samples (EPIGUT, ALLGUT) to better identify the prevalence of norovirus infections, and (3) inform future prospective studies. Samples were de-identified and assigned unique specimen numbers before laboratory testing. Charts were reviewed by an independent reviewer to collect data, including age at onset of infection, sex, diagnosis, coinfections, absolute neutrophil counts (ANCs), absolute lymphocyte counts (ALCs), and presence of norovirus symptoms, including fever, vomiting, and diarrhea. 

### 2.2. Molecular Detection of Norovirus

Stool was diluted in 1× Dulbecco’s phosphate-buffered saline (PBS) to a final concentration of 10–20% (*w*/*v*). RNA was extracted with Applied Biosystems MagMAX-96 Viral RNA Isolation Kit (Waltham, MA, USA), according to the manufacturer’s protocol, in 96-well plates loaded onto a ThermoFisher Scientific KingFisher Flex Magnetic Particle Processor (Waltham, MA, USA). For this study, infection was defined as any positive norovirus test regardless of clinical symptoms. Infections were identified by screening samples using GI- and GII-specific primers in a real-time PCR assay, as previously described [22]. Samples were considered positive for a norovirus if a cycle threshold (Ct) value was less than or equal to 38. Two positive controls and two negative controls were present on each 96-well plate. Virus-positive samples were further analyzed to determine the precise norovirus genotype by first using a set of primers specific to capsid VP1 (region D) that can amplify viruses from both GI and GII genogroups [23]. In patients that had tested positive at more than one time point, infections were classified as being distinct if they had at an intervening time point tested negative or if a different viral genogroup/genotype was detected by PCR. Chronic norovirus infections were defined as having a norovirus-positive sample beyond a 30 day period. Shedding duration was defined as the time between the first and last positive samples collected, with no negative samples in between. Samples from patients with chronic infections were examined further and both capsid and polymerase regions were amplified using RNA extracted with Qiagen RNeasy Mini Kit (Germantown, MD, USA) from 10% (*w*/*v*) stool suspensions in PBS. The capsid-encoding region of GI noroviruses was amplified with the primers GISKF/GISKR [24], and that of GII noroviruses was amplified with the primers COG2F and G2SKR [25] or G2SKF [26] and GIIR1 [27]. The polymerase region for both GI and GII noroviruses was amplified with the universal primers JV12Y and JV13I [28]. If this set of primers produced no virus-specific bands, the GI-specific GI [29] and JV13I primers and the GII-specific JV12Y and Noro II-R [29] primers were used. PCR products were purified using the Promega Wizard SV Gel and PCR Clean-Up System (Madison, WI, USA) and sequenced. Sequences were analyzed using the Norovirus Automated Genotyping Tool version 2.0 [30]. 

### 2.3. In Vitro Human Norovirus Infection

To determine human norovirus infectivity in different cell lines, BJAB, HEK293T, SY5Y, M12, RAW264.7, and Neuro-2a cells were propagated in complete medium containing 10% fetal bovine serum (FBS) and 1% penicillin/streptomycin. DMEM was used for HEK293T, M12, and RAW264.7 cells, RPMI for BJAB cells, a 1:1 mixture of EMEM and F12 medium for SY5Y cells, and EMEM for Neuro-2a cells. BJAB and M12 cells were dissociated by pipetting, HEK293T, SY5Y, and Neuro-2a cells by trypsinization, and RAW264.7 were dislodged with a cell scraper. After centrifugation and suspension in an appropriate medium, 1.3 × 10^5^ cells were inoculated with 25 µL of 1:10 (*w*/*v*) stool suspension in PBS that was positive for either GII.6 or GII.4-Sydney human norovirus. The GII.4-Sydney stool was collected by the Centers for Disease Control from a 69-year-old male who presented with symptoms during a cruise ship outbreak in January of 2013, and it has been used in a prior study reporting infection of BJAB cells [31]. The GII.6-positive stool was collected by St. Jude Children’s Research Hospital from a febrile 3.5-year-old male patient undergoing treatment for a hematologic malignancy. Samples were incubated for 2 h at 37 °C, then centrifuged at 730× *g* for 7.5 min. The supernatant was discarded, and the pellet was resuspended in 200 µL of complete medium. Half of each sample was added to the wells of a previously prepared 48-well tissue culture plate containing 900 µL of the appropriate complete medium, and the plate was incubated for 3 days at 37 °C. The remaining 100 µL of the sample was mixed with 900 µL of the appropriate complete medium and frozen at −80 °C as a sample for 0 days postinfection (dpi). To inactivate human norovirus, stool suspensions positive for GII.4 and GII.6 human norovirus were exposed to 200,000 µJ/cm^2^ of ultraviolet light 25 min before being used to infect BJAB, HEK293T, and SY5Y cell lines, as described above.

To test for infectious virus in the stools of chronically infected patients, virus-positive stool samples were thawed on ice and up to 50 mg of stool was diluted 1:10 (*w*/*v*) in ice-cold PBS. Triplicate aliquots of 1.3 × 10^5^ HEK293T cells were inoculated with each viral stool suspension (10 µL for Ct < 20, 15 µL for Ct 20–25, and 20 µL for Ct > 25, corresponding to 10^4^–10^7^ viral genome copies per inoculum) and incubated for 2 h at 37 °C. Samples were then centrifuged at 730× *g* for 7.5 min, the supernatant was carefully discarded, and the pellet was resuspended in 200 µL of complete DMEM. Half of each sample (100 µL) was added to a well of a previously prepared 48-well tissue culture plate containing 500 µL of complete DMEM, and the plate was incubated for 3 days at 37 °C. The remaining 100 µL of the sample was mixed with 500 µL of complete DMEM and frozen at −80 °C as a 0-dpi sample. 

RNA was extracted from all samples by using the Qiagen RNeasy Mini Kit (Germantown, MD, USA) in accordance with the manufacturer’s instructions. Human norovirus genome copies were enumerated as previously described by Jones et al. [32] (Method A). Experiments evaluating replication at 3 dpi were detected using Applied Biosystems QSY™-quenched RING2-TP TaqMan probe (Waltham, MA, USA) instead of a TAMRA-quenched probe. To calculate the norovirus replication efficiency for each sample, the average viral genome copy number was calculated for each triplicate infection at 0 and 3 dpi. Data are reported as the fold-increase in viral genome copy number from 0 to 3 dpi, and a sample was considered to contain infectious virus if the fold-increase was >2. 

Human intestinal enteroids derived from fetal ileum were obtained from the University of Michigan Medical School Translational Tissue Modeling Laboratory. Briefly, the 3-dimensional (3D) organoids were suspended in Corning Matrigel (Tewksbury, MA, USA) and maintained in complete L-WRN medium as described [33]. Prior to infection, 3D enteroids were dissociated into single cells and seeded as 2D monolayers in 48-well plates that were coated with human collagen type IV at a density of 1 × 10^5^ cells per well. The 2D enteroids were differentiated for 6 days, as previously described [33]. Next, unfiltered human norovirus-positive stool samples were diluted 1:10 in PBS and centrifuged for 3 min at 21,000× *g*. The supernatant was used to inoculate enteroids for 1 h at 37 °C and washed once with culture medium without growth factors. Then, 0 dpi samples were frozen immediately and fresh complete L-WRN medium was then added to the remaining samples, followed by a 3-day incubation at 37 °C with 5% CO_2_. RNA was extracted from samples using Zymo Research Direct-zol RNA MiniPrep Plus (Irvine, CA, USA) according to the manufacturer’s protocol. Human norovirus genomes were quantified by one-step RT-qPCR as described [34]. The fold-increase was determined by dividing 3 dpi viral genome copies by 0 dpi genome copies.

### 2.4. Statistical Analyses

For demographic comparisons between infected and uninfected patients, χ^2^ tests were performed to determine whether there were defining features that were significantly associated with infection status. To delineate factors associated with longer shed duration, Mann-Whitney U-tests were performed and median difference as well as 95% confidence intervals (CIs) reported. A two-sided *p*-value < 0.05 was considered statistically significant for all analyses. All statistical analyses were performed in GraphPad Prism, Version 8.2.0 (San Diego, CA, USA). 

## 3. Results

### 3.1. Cohort Description

We tested a total of 4644 available fecal samples from 1140 pediatric patients across 3 cohorts treated during the period from 2012 to 2018. There was variation in the number of samples collected each year, with a median of 720 samples collected annually (range: 143–1067). Overall, a median of 2 samples was collected per patient (range: 1–50) over a period of 1 to 1711 days (median: 48.5 days). There were slightly more male patients (*n* = 646, 57%) than female patients (*n* = 494, 43%) (Table 1). With the inclusion of patients undergoing continuous treatment beyond childhood, the patients ranged in age from less than 2 months to 29 years old, with almost half (*n* = 463, 41%) being younger than 5 years (Table 1). Most patients were white (*n* = 822, 72%) and were receiving treatment for hematologic malignancy (*n* = 660, 58%) or solid tumors (*n* = 273, 24%) (Table 1).

### 3.2. Prevalence and Temporal Trends of Norovirus Infections

From the total of 4644 samples, 214 (4.6%) tested positive for norovirus and these were collected from 123 (10.8%) of the patients in our cohort. Overall, the proportion of patients with norovirus infections mirrored the makeup of the cohort (Table 1) and there were no significant associations with sex, race, or diagnosis. While norovirus infections spanned all ages, age was significantly associated with infection status (Table 1; χ^2^ test, *p* = 0.003). We identified 2 patients who were coinfected with both GI and GII noroviruses, and 4 patients who were sequentially infected either first with a GI virus and then with a GII virus (*n* = 2) or with multiple consecutive GII viruses (2 infections: *n* = 1; 3 infections: *n* = 1). Accounting for these multiple infections, we tabulated a total of 128 infections in our cohort that were caused by 24 GI viruses (19%) and 104 GII viruses (81%). The proportions of patients with infections by either genogroup were similar based on other demographic variables, including sex, age, race, and diagnosis.

From 2012 through 2016, there was a gradual rise in the number of GII infections before they peaked in 2017 (Figure 1a). As the GII cases increased, the number of GI infections decreased after peaking in 2014 (Figure 1a). More recently, in 2018, we observed a high incidence of sample positivity, but this was largely attributed to infections in a relatively small number of patients (Figure 1a). Across all the years of study, most norovirus infections (GI and GII) occurred between September and May (Figure 1b). This suggested a prolonged season of infections in comparison to previous reports on seasonality [35,36]. 

### 3.3. Genotyping of Patient Isolates

To obtain a more detailed picture of the norovirus diversity in our patient population, we genotyped the capsid-encoding *ORF2* gene from infected patients identified in our largest cohort (FAECIS). Although we were unable to obtain a subset of genotypes due to failed sequencing reaction or mixed trace signals, we successfully genotyped 14 of the 18 GI-positive samples (77.8%) and 57 of the 131 GII-positive samples (44.3%). As genotyping was performed on more than one positive sample per patient if available, these samples collectively represent 45 virus infections from a total of 42 patients. Overall, 3 GI genotypes and 6 GII genotypes were detected (Figure 2a,b). For GI infections, GI.3 was the most prevalent, with GI.5 and GI.6 being detected in fewer cases (Figure 2a). For GII infections, GII.4 was detected in most infections (*n* = 15, 43%), followed by GII.1 viruses (*n* = 8, 23%) and GII.6 viruses (*n* = 5, 14%). GII.2, GII.3, and GII.7 were collectively detected in 7 infections (20%) (Figure 2b). We were able to characterize 3 of the 4 sequentially infected patients that were identified. Of the 2 patients with intergenotypic infections, the first was positive for GI.3 for more than 1.5 months and was then infected with a GII.6 virus 5 months later, whereas the second patient was infected with a GI virus and then, almost 3 months later, had a GII.4 infection that persisted for more than 2 months. The patient who was triply-infected with GII viruses first presented with a GII.6 virus infection lasting more than a month, which was followed by a 3-day infection with a GII.1 virus 6 months later and then a GII.1 virus infection 3 years later that lasted more than 3 months. Regarding the 2 coinfected patients, we found that the first patient was initially co-infected with GI.5 and GII viruses, whereas the second patient was infected with a GII.4 virus and then superinfected with a GI virus within 2 months. Because these mixed infections co-occurred with chronic infections, we next sought to better define this group of individuals and identify any other patients in our cohort with chronic infections. 

### 3.4. Persistent Virus Shedding in a Subset of Patients

We recorded a total of 21 chronic infections in 20 patients, representing 17% of the patients with norovirus infections. Most cases (*n* = 5–6/year) were identified in 2012–2013, whereas only 1 to 3 were identified in subsequent years of observation (Table 2). Each patient identified as a chronic shedder had between 2 and 16 samples collected longitudinally for analysis (median: 3), and, based on this sample set, we found that patients shed virus for 37 to 418 days (median: 97 days; Table 2). We next performed chart reviews to define patient and infection characteristics (Table 2 and Table 3, respectively). Chronically infected patients had a median age of 3.6 years at the onset of infection (range: 0.9–15.6 years) and comprised 9 male and 11 female patients. Seventeen patients were white, 2 were African American, and 1 had her race listed as “other.” Nineteen of the 20 chronically infected patients (95%) shared a diagnosis of hematologic malignancy, whereas the remaining patient had a solid tumor. Six of the patients underwent bone-marrow transplants during their treatment, and one of these patients developed complications related to graft-versus-host disease (Table 2). While the onset of infection occurred with varying levels of immunocompetency within patients, most were detected while the patients were immunosuppressed (*n* = 13/21, 62%), as evidenced by low ANCs (<1500/mm^3^). In rare cases (*n* = 4), these counts improved at subsequent time points, at which the patients continued to shed virus. ALCs were low (<2000/mm^3^) for all patients throughout their infections. 

All but two patients displayed symptoms of gastrointestinal illness (i.e., fever, diarrhea, and vomiting) during at least one time point. The most common symptom during infection was diarrhea (*n* = 16/21 infections, 76%) followed by fever (*n* = 15/21 infections, 71%) and then vomiting (*n* = 9/21 infections, 43%). These symptoms occurred both throughout and intermittently during the chronic norovirus infections. However, most patients’ infections (*n* = 16) co-occurred with another pathogen, making the ability to attribute symptoms to norovirus alone difficult. *Clostridioides difficile* (*n* = 7) and adenovirus (*n* = 4) were the most frequently detected coinfections, whereas enteric viruses, including astrovirus and rotavirus, as well as respiratory pathogens, including human rhinovirus, enterovirus, and parainfluenza virus, were detected to lesser extents. Infections with other viruses (cytomegalovirus, BK virus, and Epstein-Barr virus), bacteria (*Klebsiella pneumoniae* and *Staphylococcus epidermidis*), or fungi (*Candida parapsilosis*) were also detected in a subset of patients. Fourteen of the 21 infections were confirmed as newly acquired based on available samples that were collected previously and tested negative for norovirus (Table 3). We also found that 7 patients eventually cleared their norovirus infections, as evidenced by subsequent negative samples.

Additional genotyping of both *ORF1* (encoding the polymerase) and *ORF2* (encoding the capsid) was performed on samples from chronic shedders to better define their viral infection(s) (Table 3). Overall, out of 48 stool samples collected from chronic virus shedders that were submitted for genotyping, only 3 (6.3%) did not yield specific PCR products. Confirmation of each of the following genotypes was achieved with multiple samples per patient, except for Patients 16 and 19, where only one sample could be successfully sequenced. Eleven shedders were infected by the predominant epidemic GII.4 Sydney 2012 norovirus genotype in combination with the following *ORF1* genotypes: 4 with the GII.Pe genotype, 3 with the GII.Pg genotype, and 1 each with the GII.P4, GII.P4 New Orleans 2009, and GII.P16/GII.4 genotypes. The patient infected with the GII.P4 genotype was later superinfected with a GI.5 virus. The polymerase sequence was not determined for one of the GII.4 Sydney 2012-infected shedders (Patient 8). One patient was infected with the GII.4 Den Haag 2006b that was found in combination with the GII.P4 genotype, and this patient was also coinfected with GI.5 virus. The second most prevalent virus in the cohort was the GII.6 genotype (*n* = 3), with one patient being infected with GII.P16/GII.6 and one with GII.Pg/GII.16. The sequence of the polymerase gene from the third patient could not be determined. Four shedders were infected with other GII genotypes, including GII.Pe/GII.1, GII.P16/GII.2, GII.P16/GII.3, and a GII.P16/GII.1 virus (this patient also had GII.Pg detected at a different time point). Two chronic shedders were infected with GI.P3/GI.3 noroviruses. We next explored whether certain virus or host factors were associated with longer shedding times. Most patients infected with GI and GII viruses shed for less than 100 days; however, a subset of GII-infected patients shed for much longer (Figure 3a). With approximately half of the cohort being infected with GII.4 viruses, we compared the duration of shedding in these patients and in all other chronic shedders but found no significant difference (median difference = −25.5; 95% CI = −71, 66; *p* = 0.768). We next examined whether there were associations between shedding duration and demographic or clinical variables. We found that male patients shed virus for longer when compared to female patients (median difference = −121; 95% CI = −151, −2; *p* = 0.043). Also, if we dichotomized patients on whether or not diarrheal symptoms were reported, we found that diarrhea was associated with longer shed duration (median difference = −71.5; 95% CI = −148, −6; *p* = 0.024; Figure 3b). For comparison, we dichotomized patients according to vomiting symptoms and did not find the same association with shed time (median difference = −6; 95% CI = −60, 85; *p* = 0.82; Figure 3b). For these analyses, patients with diarrhea could have also had vomiting or fever at the same time, and vice versa. We detected no significant associations between age, diagnosis, ANC levels (as a proxy for immune status), coinfections, or Ct value (as a proxy for virus load) and shedding time.

### 3.5. Infectivity of Noroviruses in Chronic Shedders

We recently discovered serendipitously that HEK293T cells support human norovirus replication at levels comparable to those in BJAB B cells when infected with GII.6- or GII.4-positive stool inocula collected from acutely infected patients (Figure 4a). Because HEK293T cells are neuronal-like [37], we investigated whether other neuronal cells would also support viral replication. Indeed, viral genome copy number increases were comparable in SY5Y cells (Figure 4a). To bolster our finding that these cell lines support human norovirus replication, we tested a panel of murine cell lines that we anticipated to be resistant to human norovirus infection, considering the strict species specificity of noroviruses. Consistent with this, we observed no increase in viral genome copy numbers when murine M12 B cells, RAW264.7 macrophages, or Neuro-2A neuronal cell lines were inoculated with the same GII.6-positive stool (Figure 4a). Furthermore, ultraviolet treatment of GII.6- and GII.4-positive acute stools completely ablated virus infectivity in BJAB, HEK293T, and SY5Y cells (Figure 4b). 

Having established that HEK293T cells are susceptible to human norovirus infection, we next used this in vitro system to test the infectivity of samples from each of the chronic shedders identified in our cohort (*n* = 62 total samples). Overall, the virus was propagated in 49 samples (75%; Figure 5a), which translated to 16 of 20 patients having infectious virus in their stool at one or more time points beyond the start of their infections (Figure 5a). Notably, 15 of these 16 patients shed infectious virus for more than 30 days beyond the initial sample collection. Although our sample size is insufficient to permit testing for statistical differences between genogroups and genotypes, it is noteworthy that a wide spectrum of human noroviruses, including GI.P3/GI.3 and various GII genotypes, could establish chronic infections associated with the shedding of infectious virus (Figure 5b). 

Because intestinal enteroids also support human norovirus infection [8,38], we tested a subset of samples (*n* = 7) from the chronic shedders that replicated most efficiently in HEK293T cells for infectivity in enteroids. Data are reported as fold-increase of viral genome copy numbers over a 3-day infection (Figure 6). A range of patterns was observed. Three of the 7 patient samples did not replicate efficiently in enteroids (<5-fold increase in genome copies), whereas 3 other samples replicated more efficiently in enteroids than in HEK293T cells (>100-fold increase in genome copies). The 7th sample replicated to comparable levels in both in vitro systems. Together, these data (i) support our findings in HEK293T cells that infectious human norovirus can be shed from chronically infected patients, (ii) confirm prior studies that demonstrated variability in replication using the enteroid model [38,39], and (iii) reveal virus- or stool-dependent variability in replication in different cell types.

## 4. Discussion

Noroviruses are a leading cause of acute diarrhea worldwide, with immunocompromised patients being at risk for chronic infections [17,18,40,41]. Over a 6-year period (2012–2018), we investigated norovirus infections in immunocompromised patients undergoing treatment for cancer, immunodeficiencies, or infectious diseases. We found a 10.8% prevalence of norovirus during this time period, with the peaks in 2014 and 2017 for GI and GII viruses, respectively. We further identified 20 patients experiencing chronic norovirus infections, including 6 with dual or sequential infections. Critically, by using a novel HEK293T and an established enteroid in vitro system, we showed that virus detected in samples during the chronic phase of infection was infectious. Together, these data highlight the importance of studying chronic norovirus infections among immunocompromised hosts, as prolonged shedding of infectious virus could impact the likelihood of transmission and risk of nosocomial infections. 

Our data indicate that noroviruses represent a significant enteric virus burden for our immunocompromised patient population. The patient prevalence observed in this study is comparable to that previously reported in immunocompromised cohorts (11%) [42], as is the viral diversity, which included 10 genotypes but was dominated by GII.4 and GII.6 viruses. Although the proportion of chronic shedders we identified (16.4%) was lower than that in previous reports (33–57%) [18,42], this is the largest cohort studied to date by retrospective analysis, and our 6-year window of observation allowed us to capture sequential infections. We observed temporal trends that were consistent with prior studies that showed more infections occurring during the first quarter of the year when air temperatures are lower [43], but we also noted that >2% of patients experienced norovirus infections from September all the way until May, indicating an extended season of norovirus infections in our patient population. It is interesting to consider to what extent this supports the idea that immunocompromised populations could act as a viral reservoir for emergent strains [9,44,45], particularly those that occur off-season, or whether these infections merely represent introductions of viruses from the community [44,46]. Future longitudinal studies of norovirus infections in large cohorts of both immunocompromised and healthy individuals within a community would help shed light on this important question. Finally, closer examination of patients with chronic shedding revealed a sex-based difference in shedding duration. Although we could not identify measurable differences in age, diagnosis, coinfections, viral factors, or immune status that could explain the differences in shed time between males and females, this does not rule out some other confounding factor that we did not capture in our dataset. While this difference has not been reported previously for norovirus, our finding is supported by previous studies of other viral infections in animal models [47,48]. 

Although it is well-documented that norovirus-infected immunocompromised individuals can continue to shed viral genomic RNA in their stool for months or even years [7,18,40,41], the lack of human norovirus culture systems has historically precluded testing of whether these patients shed infectious virus. We and others have recently reported that human noroviruses can replicate modestly in B-cells and enteroids [8,31,32,38,39], enabling us to address this gap in our knowledge. Our discovery that HEK293T cells are capable of supporting virus infection represents a quick, standardized and low-cost system to evaluate the infectivity of patient samples. While additional head-to-head studies are needed to fully characterize the benefits and limitations of this new in vitro system, using this method, we found that the shed virus remained infectious in 76% of chronic cases. Such a high percentage of patients with prolonged shedding of infectious virus underscores the importance of a previous report that showed epidemiologically linked transmissions from three chronic shedders within a hospital [49]. Because we also found that diarrheal symptoms were associated with longer shedding duration, this translates to the potential need for extended durations of isolation precautions for infected immunocompromised patients.

Although our study has many strengths, it is not without limitations. First, the retrospective nature of our study precluded us from obtaining symptom information at the precise onset. Rather, we relied on chart reviews, which may or may not have clearly defined subjective symptoms such as diarrhea. Also, our patient population experiences sporadic symptoms (i.e., fever, diarrhea, vomiting) as a result of their treatment as well as numerous coinfections with gastrointestinal pathogens, clouding our ability to remark on whether symptoms may be informative for chronic infections and hospital infection control. We were also unable to accurately describe the duration of shedding for many of the chronically infected patients (Table 3), in addition to the greater cohort, because these individuals remained norovirus-positive at the final time point available for testing. Therefore, we may be missing the precise duration for chronic shedding as well as the inclusion of additional chronic shedders. Thus, it is possible that these individuals shed virus for a longer period of time than we have reported. Secondly, although we were frequently able to confirm genotypes across multiple sample patients, we could obtain genotype data (for either the capsid gene alone or for both the capsid and polymerase genes) only from a subset of the total infections (*n* = 51, 40%). Despite our efforts using multiple genotyping methods, questions remain regarding the “true” prevalence of the diverse norovirus strains circulating in our patient population. Additionally, we were limited in our ability to fully resolve the within-host viral dynamics of dual-infected patients. In particular, Patient 17 appeared to have 3 sequential infections, 2 of which were chronic. Without intervening samples collected between 2 of these infections, we cannot be certain that they were not mixed infections or the result of superinfections. However, we detected 2 polymerase genes and 1 capsid gene (Table 3) during the third infection, raising the possibility that the 2 infections may have overlapped. Deep sequencing at the various time points would be required to further elucidate this possibility. Still, our data describing both coinfections and sequential infections in the context of prolonged virus shedding could lend support to the idea that immunocompromised individuals are a source of emergent strains [9,49] and additional studies are greatly needed. 

In conclusion, our data support the current literature on norovirus infections in immunocompromised hosts by demonstrating that persistently shed virus remains infectious and the current guidelines for isolation precautions should be re-evaluated to reduce the potential for transmission in hospital settings.

## Figures and Tables

**Figure 1 viruses-12-00619-f001:**
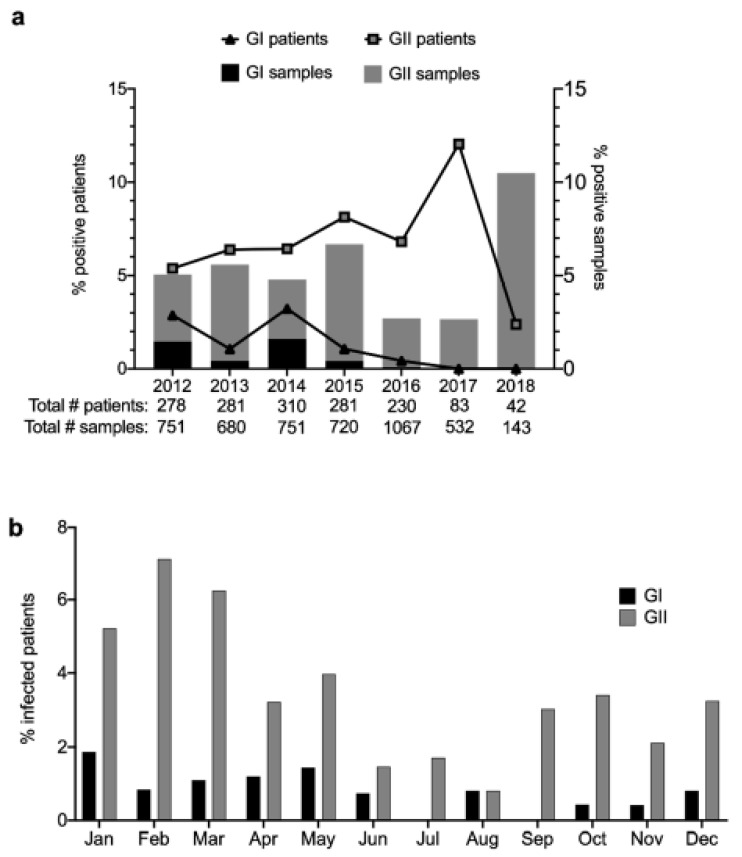
Human norovirus GI and GII infections identified from fecal samples collected over a 6-year period (2012–2018) at a pediatric research hospital. (**a**) Sample and patient positivity are noted for each year, with total numbers of samples and patients analyzed each year listed below (**b**) Seasonal trends observed over time indicate that the burden of norovirus infections was highest from September to May.

**Figure 2 viruses-12-00619-f002:**
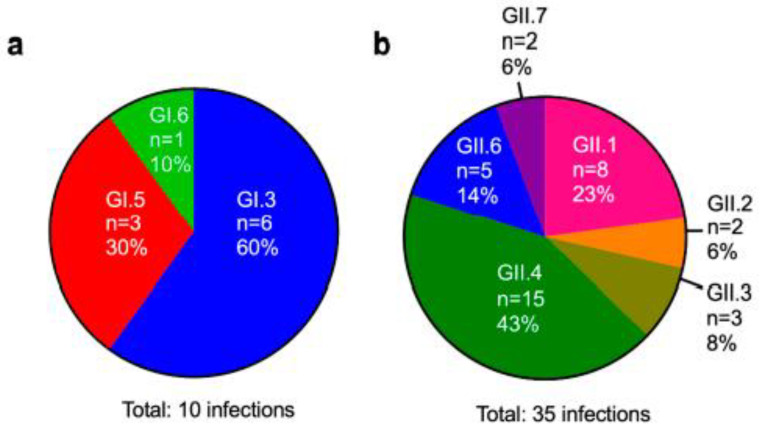
Co-circulating human norovirus strains in immunocompromised patients at a pediatric research hospital. Three GI genotypes (**a**) and 6 GII genotypes (**b**) were detected with varying frequencies.

**Figure 3 viruses-12-00619-f003:**
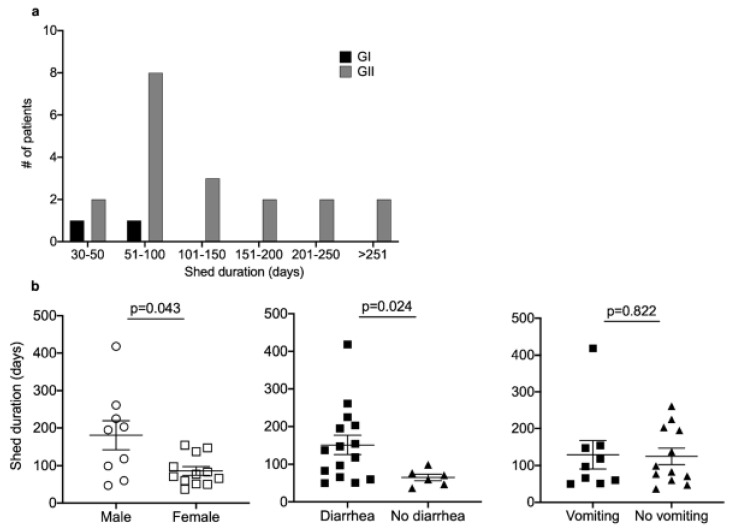
A subset of 21 infections was characterized by chronic virus shedding for >30 days. The number of patients and duration of virus shedding (**a**) varied for GI and GII infections, but most patients shed for <100 days. (**b**) An exploratory analysis of variables associated with longer shedding implicated male sex and the presence of diarrheal symptoms, but not vomiting, as significant. Statistical difference between groups was determined by the Mann-Whitney U-test, with a 2-sided *p*-value of <0.05 denoting significance.

**Figure 4 viruses-12-00619-f004:**
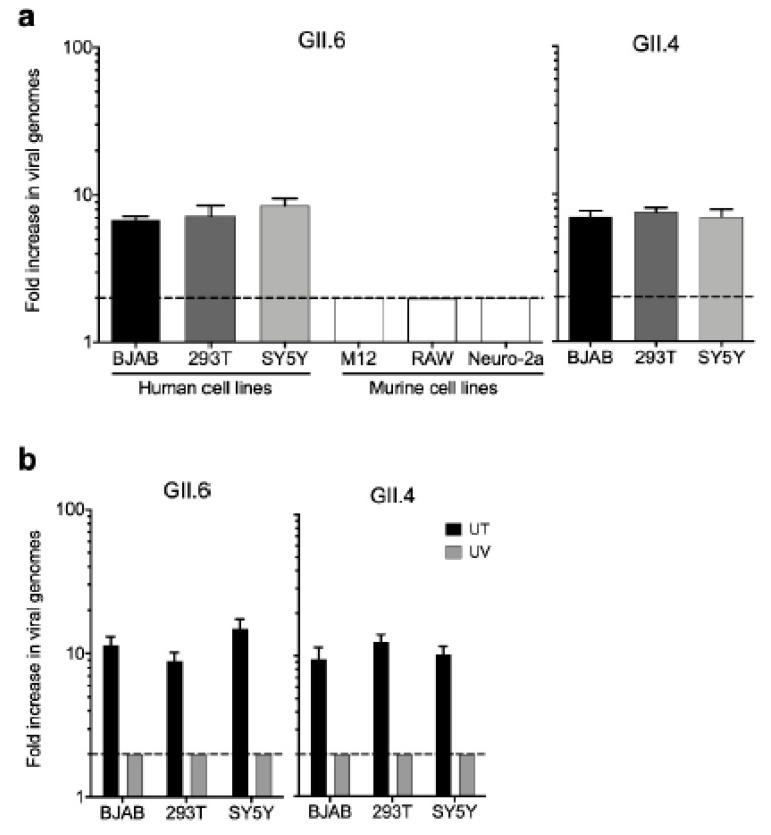
HEK293T cells support human norovirus replication. (**a**) Human norovirus genotypes GII.6 and GII.4 replicate in human but not murine cell lines. (**b**) Ultraviolet (UV) treatment inhibits the replication of GII.6 and GII.4 noroviruses in 3 human cell lines (UT = untreated). The dashed line represents the lower limit of detection. Error bars indicate standard error of the mean (SEM) values. Data represent the averages from 3 independent experiments.

**Figure 5 viruses-12-00619-f005:**
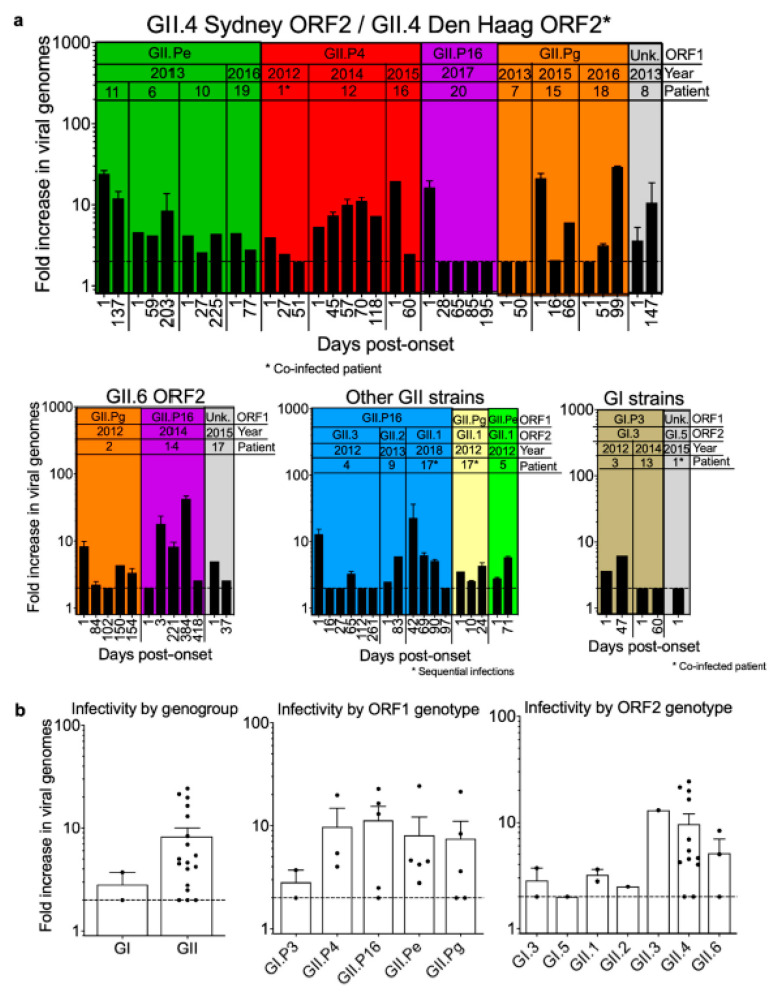
Infectious virus is consistently detected in fecal samples from chronically infected patients. (**a**) Levels of infectious virus detected in HEK293T cells inoculated with human norovirus genotypes, according to capsid (*ORF2*) and polymerase (*ORF1*) genotypes. The shaded boxes indicate samples with the same polymerase (*ORF1*) genotype. Three patients (Patients 7, 12, and 1) did not have detectable infectious virus in any of the samples tested, and one patient did not have any after the initial sample collection. (**b**) Infectivity of different noroviruses subdivided by GI and GII genogroups and by different *ORF1* and *ORF2* genotypes. The dashed line represents the lower limit of detection. Error bars indicate SEM values. Data represent the averages from 2 independent experiments.

**Figure 6 viruses-12-00619-f006:**
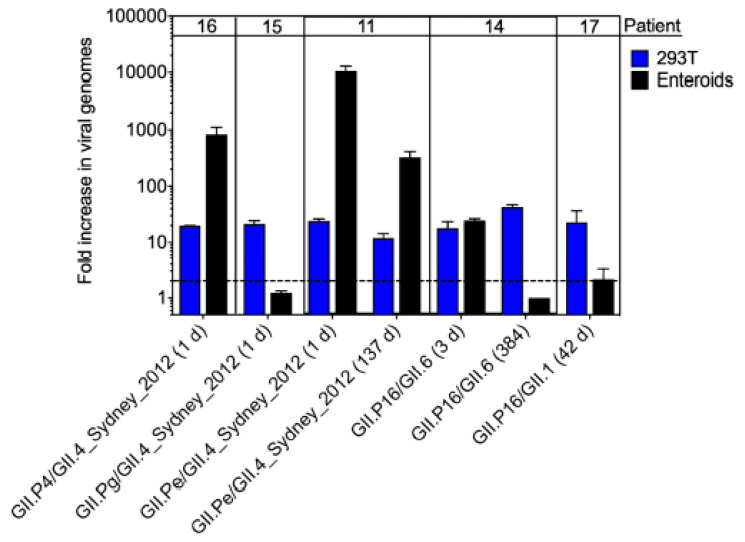
Human norovirus in stool samples from chronic shedders is infectious in human intestinal enteroids. 2D fetal ileum enteroids were differentiated for 6 days and then infected with unfiltered human norovirus-positive stools for 3 days. Viral genome copy numbers were measured at 0 and 3 dpi by RT-qPCR. Each sample was tested in triplicate and the entire experiment repeated twice. The fold-increase in viral titer was determined by comparing genome titers at 3 to 0 dpi. Replication data in HEK293T cells for each stool sample tested in enteroids are shown for comparison. The dashed line represents the lower limit of detection. Values for technical replicates were averaged and error bars indicate the mean of experimental repeats.

**Table 1 viruses-12-00619-t001:** Cohort characteristics.

	Total Patients (*n* = 1140)		Infected Patients (*n* = 123)		
	n	%	N	%	*p*-Value
**Sex**					**0.893**
Male	646	57	69	56	
Female	494	43	54	44	
**Age** (**years**)					**0.003**
0–5	463	41	69	56	
6–10	249	22	17	14	
11–15	214	19	22	18	
16–20	175	15	11	9	
21+	39	3	4	3	
**Race**					**0.712**
White	822	72	95	77	
Black	222	19	18	15	
Asian	25	2	3	2	
Multiple	63	6	6	5	
Other	7	1	1	1	
Unknown	1	0	0	-	
**Diagnosis**					**0.064**
Hematologic Malignancy	660	58	87	71	
Solid Tumor	273	24	21	17	
Brain Tumor	102	9	8	7	
Hematologic Disorder	52	5	4	3	
Infectious Disease	35	3	1	1	
Primary Immunodeficiency	18	2	2	2	

**Table 2 viruses-12-00619-t002:** Characteristics of patients with chronic norovirus shedding.

Patient	Year	Days Shed	Age at Onset (Yrs)	Sex	Race	Diagnosis	BMT	GVHD	ANC ^#^ at Onset	ANC ^#^ during Infection	Fever at Onset	No. Time Points with Fever	Diarrhea at Onset	No. Time Points with Diarrhea	Vomiting at Onset	No. Time Points with Vomiting
1	2012	51	2.9	Female	Other	HM	N	-	Low	Low	N	0/3	Y	3/3	N	1/3
2	2012	154	14.0	Female	White	HM	N	-	Normal	Normal/Low	N	4/8	Y	5/8	Y	5/8
3	2012	47	1.7	Male	White	HM	Y	N	Low	Low	N	0/2	N	0/2	N	0/2
4	2012	261	14.1	Male	White	HM	N	-	Normal	Normal/Low	N	0/6	Y	6/6	N	0/6
5	2012	71	6.8	Female	White	HM	N	-	Normal	Low	N	1/2	N	1/2	N	0/2
6	2013	203	2.3	Male	White	HM	N	-	Low	Low	N	1/3	Y	2/3	N	0/3
7	2013	50	15.6	Female	Black	HM	Y	N	Low	Low	N	0/2	Y	2/2	Y	1/2
8	2013	147	5.1	Female	Black	HM	N	-	Normal	Normal	Y	3/4	N	1/4	N	1/4
9	2013	83	1.9	Female	White	HM	N	-	Low	Low/Normal	Y	2/2	N	1/2	N	0/2
10	2013	225	2.1	Male	White	HM	N	-	NR	Normal	N	1/3	Y	3/3	N	0/3
11	2013	137	6.4	Female	White	HM	N	-	Low	Low	Y	2/2	Y	1/2	N	0/2
12	2014	118	11.6	Male	White	HM	N	N	Low	Low	N	3/11	Y	9/11	Y	8/11
13	2014	60	1.2	Male	White	HM	N	-	Normal	NR	N	0/1 ^+^	N	0/1 ^+^	N	0/1 ^+^
14	2014	418	1.0	Male	White	HM	N	-	Low	Normal	N	2/7	Y	6/7	Y	5/7
15	2015	66	13.0	Female	White	HM	N	-	Low	Low	N	1/3	Y	1/3	Y	1/3
16	2015	60	3.5	Female	White	HM	N	-	Low	Low	N	1/3	Y	2/3	Y	1/3
17	2015	37	1.8	Female	White	HM	N	-	Normal	Low	N	1/2	N	0/2	N	0/2
18	2016	99	10.1	Male	White	ST	Y	Y	High	High/Normal	Y	1/7	N	0/7	N	0/7
19	2016	77	3.6	Female	White	HM	N	-	Low	Low	N	0/2	N	0/2	N	0/2
20	2017	195	2.1	Male	White	HM	Y	N	Low	Low/Normal	N	2/5	N	1/5	N	0/5
17	2018	97	4.1	Female	White	HM	Y	-	Low	Low/Normal	N	1/14	N	11/14	N	2/14

HM, hematologic malignancy; ST, solid tumor; NR, not recorded; **^#^** Absolute neutrophil count, low (<1500/mm^3^), high (>8000/mm^3^); ^+^ No medical record available for the second time point.

**Table 3 viruses-12-00619-t003:** Infection characteristics of patients with chronic shedding.

Patient	Infection Classification	New Infection	Cleared Infection	Coinfections Other Pathogens	Genotype (*ORF1*/*ORF2*)	Infectious Virus Detected
1	Coinfection	Unknown	Unknown	None	/GI.5 and GII.P4/GII.4 Den Haag 2006b	Y
2	Single	Y	Y	*C. difficile*, astrovirus	GII.Pg/GII.6	Y
3	Sequential ^&^	Unknown	Y	*C. difficile*	GI.P3/GI.3	Y
4	Single	Unknown	Unknown	*C. difficile*	GII.P16/GII.3	Y
5	Single	Unknown	Unknown	Adenovirus	GII.Pe/GII.1	Y
6	Single	Y	Unknown	Rotavirus	GII.Pe/GII.4 Sydney 2012	Y
7	Single	Y	Y	CMV, BK Virus	GII.Pg/GII.4 Sydney 2012	N
8	Single	Unknown	Y	None	/GII.4 Sydney 2012	Y
9	Single	Y	Unknown	*C. difficile*	GII.P16/GII.2	Y
10	Single	Y	Unknown	*C. difficile*, Epstein-Barr virus	GII.Pe/GII.4 Sydney 2012	Y
11	Single	Unknown	Unknown	None	GII.Pe/GII.4 Sydney 2012	Y
12	Single	Y	Unknown	*K. pneumoniae*, *S. epidermidis*, BK virus, adenovirus, parainfluenza virus	GII.P4 New Orleans 2009/GII.4 Sydney 2012	Y
13	Single	Y	Unknown	None	GI.P3/GI.3	Y
14	Single	Y	Unknown	Adenovirus, parainfluenza virus	GII.P16/GII.6	Y
15	Sequential ^$^	Unknown	Y	*C. difficile*	GII.Pg/GII.4 Sydney 2012	Y
16	Superinfection *	Y	Y	Human rhinovirus, enterovirus	GII.P4/GII.4 Sydney 2012	Y
17	Sequential ^#^	Y	Y	Human rhinovirus, enterovirus	/GII.6	Y
18	Sequential ^@^	Y	Unknown	*C. parapsilosis*, adenovirus	GII.Pg/GII.4 Sydney 2012	Y
19	Single	Y	Unknown	None	GII.Pe/GII.4 Sydney 2012	Y
20	Single	Y	Unknown	*C. difficile*, astrovirus	GII.P16/GII.4 Sydney 2012	Y
17	Sequential ^	Y	Unknown	None	GII.Pg and GII.P16/GII.1	Y

^&^ Subsequently infected with a GII.6 virus 5.4 months later; ^$^ Previously infected with GI virus 2.8 months prior; * Became superinfected with GI infection; ^#^ Subsequently infected with an acute GII.1 virus infection 6.6 months later, followed by a chronic GII.1 virus infection almost 1.7 years later in 2018, see ^; ^@^ Previously infected with a GII virus 2.2 years prior. ^ Previously infected with the GII.6 virus 1.7 years prior in 2015, see ^#^.

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
