# Peer review of "Infectious Norovirus Is Chronically Shed by Immunocompromised Pediatric Hosts"

_viruses, 2020, doi:10.3390/v12060619_

Round 1
Reviewer 1 Report
The paper is well written and thought provoking. It confirms that chronic sheders are infectious. perhaps more thought provoking is the identification of a new HuNoV replication system. I recommend its publication once some minor deficiencies are addressed.
the title could be stronger...something that reflects the younger age bias of the study. Perhaps "Infectious norovirus is chronically shed by immunocompromised pediatric patients"?
Title page: #Authors contributed?? (equally?)
line 43: norovirus as a cause of severe diarrhea...seems a little hyperbolus
the first paragraph of section 2.3 is way too cryptic. I am not sure I could repeat the work from the description; particularly the inoculation procedure- cells were trypsinized and free floating? How were they propagated?...media types? What is "appropriate media?" what was the stool used to determine the tissue culture method adult of child? any references? perhaps manuscripts in press or in preparation?
As I am sure the authors are aware the Estes group reports that norovirus replicates better in vitro when the stool source is from younger patients. This point is not really raised in the discussion or elsewhere in the paper. Does this play a role in the results reported here?
Figure 3a: something indicating "days post infection" or maybe "shedding duration" on the x axis would clarify this figure.
In Fig 4a and 4b these result are clearly different experiments but the results are also a little different; were these different inocula or the same source?
in Fig 5 there is a considerable increase over replication observed in fig 4 please comment as to why?
Fig 6 will be confusing to the reader, at least I was initially. Readers will think that 6a and 6 b are separate incongruent results until they recognize that 0 dpi represents inoculum in 6a. I would recommend not showing fig 6a; or perhaps clarifying the figure legend to prevent confusion. also just an FYI, the enteriod system requiresa higher inoculum than 1.5 log to work reliably. I think the Estes group recommends a three log inoculum.
Author Response
The paper is well written and thought provoking. It confirms that chronic sheders are infectious. perhaps more thought provoking is the identification of a new HuNoV replication system. I recommend its publication once some minor deficiencies are addressed.
Response: We appreciate the reviewer’s interest in our study and recommendation for publication.
the title could be stronger...something that reflects the younger age bias of the study. Perhaps "Infectious norovirus is chronically shed by immunocompromised pediatric patients"?
Response: We have taken the reviewers comment into consideration and edited the description of our population within the title.
Title page: #Authors contributed?? (equally?)
Response: We thank you for bringing this typographical error to our attention. The term “equally” has been added.
line 43: norovirus as a cause of severe diarrhea...seems a little hyperbolus
Response: We have edited the sentence with the reviewer’s advice.
the first paragraph of section 2.3 is way too cryptic. I am not sure I could repeat the work from the description; particularly the inoculation procedure- cells were trypsinized and free floating? How were they propagated?...media types? What is "appropriate media?" what was the stool used to determine the tissue culture method adult of child? any references? perhaps manuscripts in press or in preparation?
Response: We apologize for not providing sufficient detail in this paragraph. We have now provided more detail.
As I am sure the authors are aware the Estes group reports that norovirus replicates better in vitro when the stool source is from younger patients. This point is not really raised in the discussion or elsewhere in the paper. Does this play a role in the results reported here?
Response: Because our entire cohort was pediatric, we cannot conclude whether the age of the stool donor impacted infectivity.
Figure 3a: something indicating "days post infection" or maybe "shedding duration" on the x axis would clarify this figure.
Response: Thank you for highlighting a missing detail. We have added a title to the x-axis of figure 3a.
In Fig 4a and 4b these result are clearly different experiments but the results are also a little different; were these different inocula or the same source?
Response: These were the same source of GII.6 and GII.4 viruses.
in Fig 5 there is a considerable increase over replication observed in fig 4 please comment as to why?
Response: The source of virus for infections performed in Figs. 4 and 5 are different. There is variability in the efficiency of human norovirus replication in other systems reported to date (BJABs, enteroids) so it is not surprising that this is the case in HEK293T cells as well. We highlight this variability in the text.
Fig 6 will be confusing to the reader, at least I was initially. Readers will think that 6a and 6 b are separate incongruent results until they recognize that 0 dpi represents inoculum in 6a. I would recommend not showing fig 6a; or perhaps clarifying the figure legend to prevent confusion. also just an FYI, the enteriod system requiresa higher inoculum than 1.5 log to work reliably. I think the Estes group recommends a three log inoculum.
Response: As recommended, we have removed Fig. 6a. We can see where this would be confusing so we thank the reviewer for this suggestion.
Reviewer 2 Report
Norovirus is the major cause of acute gastroenteritis in all age-groups. In healthy individuals norovirus infections are resolved with days or weeks, however, immunocompromised individuals can be chronically infected for months or years. The extreme variability presented by norovirus in immunocompromised patients led to the hypothesis that immunocompromised patients could be the reservoirs of novel noroviruses (Vega et al. Journal of Virology 2014; Karst and Baric Journal of Virology 2015); however, there is no empirical evidence that during the chronic phase immunocompromised patients could infect healthy individuals. In this manuscript Davis et al. present evidence that chronically-infected immunocompromised patients shed infectious norovirus. Moreover, the authors suggest that norovirus can replicate in neuronal-like cells, HEK293T and SY5Y, however not enough evidence was provided to support this claim. This reviewer believes that the authors should concentrate on reporting the epidemiological data and the presence of infectious virus in immunocompromised patients by using the established enteroids system. The use of HEK293T and SY5Y cells for norovirus in vitro cultivation could be presented in a follow-up study after additional evidence is acquired.
Major points:
1. While a large epidemiological study is presented in this manuscript the most relevant data of this study is the presence of infectious virus in chronically-infected immunocompromised patients. To proof this, authors used the -controversial- BJAB and the enteroids system to cultivate norovirus. Moreover, the authors claim that serendipitously discovered that noroviruses can replicate (at high levels, line 335) in human embryonic kidney 293 (HEK293T) and SY5Y cells. The only data supporting replication in HEK293T and SY5Y is a modest 10 fold increase in viral replication. Authors did not provide any evidence on the production of viral proteins or novel virions upon inoculation with stool samples. The laboratories collaborating in this study have all reagents needed to properly prove replication on those cells. Do these cells express HBGA carbohydrates on the membrane, and what is the FUT2 gene status of these cells? This information is important as both are markers of susceptibility for human noroviruses (Nordgren and Svensson Viruses 2019).
2. The BJAB system to cultivate norovirus has been controversial due to multiple factors, one of them it is the biological significance of B cells in norovirus infection in humans (Jones et al. Nature Protocols 2015; Karst and Wobus PLoS Pathogens 2015; Brown et al. Clinical Infectious Diseases 2016). Authors have analyzed hundreds of immunocompromised patients from 3 studies. It would be interesting to know norovirus infection status (e.g. positivity, duration of shedding) of the B-cell-deficient patients from these cohorts.
3. Norovirus frequency on immunocompromised patients presented a seasonality that coincides with that of the community. This suggest that immunocompromised patients are getting the virus circulating in the community and not vice versa as suggested on this and other studies (Karst and Baric Journal of Virology 2015; Vega et al. Journal of Virology 2014). Genetic evidence support the former (Bok et al. Open Forum Infectious Diseases 2016; van Beek et al. Journal of Infectious Diseases 2017). Please review your genetic data and discuss the findings.
4. One major drawback of the study is that it seems (not specified in Materials and Methods sections) that the fold increase of the viral RNA was calculated without using a viral RNA standard for the qRT-PCR. Please provide additional details on how the fold increase was calculated and normalized among the different samples. Notably, approximately 70% of the viruses that authors claim to present viral replication have <10 fold increase. I wonder how authors account for the inherent variability of the assay in stool samples. Please provide evidence on the reproducibility of your detection/quantification method.
Minor points:
1. Line 61: Please correct the statement "Norovirus infections in healthy individuals resolve themselves in a few days [14]." While symptoms can be resolved within days, norovirus shedding has been shown to persist for weeks in healthy individuals (e.g. Atmar et al. Emerging Infectious Diseases 2008; Parra and Green Emerging Infectious Diseases 2014).
2. Please correct the statement "Until recently, a new GII.4 strain emerged every 2–4 years [9,10]." While that statement is correct for certain strains, the US95 variant emerged in 1995 (Vinje et al. Journal of Infectious Diseases 1997) and was replaced seven years later by the Farmington Hills variant (Lopman et al. Lancet 2004).
3. Please comment the reasons that the genotype of only 44% (57/131) of the GII-positive samples analyzed was identified.
4. Figure 3: Please note that x-axis legend should read "days."
5. Did the authors made any attempt to adapt the viruses collected from immunocompromised patients?
Author Response
Norovirus is the major cause of acute gastroenteritis in all age-groups. In healthy individuals norovirus infections are resolved with days or weeks, however, immunocompromised individuals can be chronically infected for months or years. The extreme variability presented by norovirus in immunocompromised patients led to the hypothesis that immunocompromised patients could be the reservoirs of novel noroviruses (Vega et al. Journal of Virology 2014; Karst and Baric Journal of Virology 2015); however, there is no empirical evidence that during the chronic phase immunocompromised patients could infect healthy individuals. In this manuscript Davis et al. present evidence that chronically-infected immunocompromised patients shed infectious norovirus. Moreover, the authors suggest that norovirus can replicate in neuronal-like cells, HEK293T and SY5Y, however not enough evidence was provided to support this claim. This reviewer believes that the authors should concentrate on reporting the epidemiological data and the presence of infectious virus in immunocompromised patients by using the established enteroids system. The use of HEK293T and SY5Y cells for norovirus in vitro cultivation could be presented in a follow-up study after additional evidence is acquired.
Response: While we agree that a more detailed follow-up study of human norovirus replication in HEK293T and SY5Y cells is warranted, we do not think that should negate the inclusion of these data here. We are using the new system here as a tool to determine whether virus in stool of chronically infected patients is infectious so showing increased viral genome copy number addresses our goal. Our conclusion that increased genome copy number reflects viral replication in these cells is supported by multiple controls: 1) no increase in viral genome copy number was observed when stools were inoculated onto murine cells; 2) UV treatment abolished infectivity of the virus in these cells; and 3) a subset of stools shown to contain infectious virus in HEK293T cells were shown to be infectious in the established enteroid system as well.
Major points:
- While a large epidemiological study is presented in this manuscript the most relevant data of this study is the presence of infectious virus in chronically-infected immunocompromised patients. To proof this, authors used the -controversial- BJAB and the enteroids system to cultivate norovirus. Moreover, the authors claim that serendipitously discovered that noroviruses can replicate (at high levels, line 335) in human embryonic kidney 293 (HEK293T) and SY5Y cells. The only data supporting replication in HEK293T and SY5Y is a modest 10 fold increase in viral replication. Authors did not provide any evidence on the production of viral proteins or novel virions upon inoculation with stool samples. The laboratories collaborating in this study have all reagents needed to properly prove replication on those cells. Do these cells express HBGA carbohydrates on the membrane, and what is the FUT2 gene status of these cells? This information is important as both are markers of susceptibility for human noroviruses (Nordgren and Svensson Viruses 2019).
Response: We did not intend to claim that HEK293T cells support high levels of viral replication and have removed that phrase. In fact, none of the available systems support particularly robust levels of replication nor have they been reported to support continued passaging of the virus. We were careful to describe increases in viral genome copy number as evidence that the virus can replicate in these cells but did not describe this as a propagation system because, like in BJABs and enteroids, we could not continually passage virus from these cultures. HEK293T cells do not express functional FUT2 (PMID: 29467317), but we inoculated them with unfiltered virus-positive stool as has been done previously in BJABs. In this case, commensal bacteria in the stool samples provide the necessary HBGA (PMID: 25378626).
- The BJAB system to cultivate norovirus has been controversial due to multiple factors, one of them it is the biological significance of B cells in norovirus infection in humans (Jones et al. Nature Protocols 2015; Karst and Wobus PLoS Pathogens 2015; Brown et al. Clinical Infectious Diseases 2016). Authors have analyzed hundreds of immunocompromised patients from 3 studies. It would be interesting to know norovirus infection status (e.g. positivity, duration of shedding) of the B-cell-deficient patients from these cohorts.
Response: We respectfully disagree that B cell infection by noroviruses is controversial. It may not be a universal feature of every norovirus infection but multiple immune cell types including B cells have been identified as targets of murine norovirus in mice (PMID: 29109476), human norovirus in chimpanzees (PMID: 21173246), human norovirus in miniature piglets (PMID: 29106738), and human norovirus in post-weaning gnotobiotic pigs (PMID: 30661347). Moreover, while Brown et al. reported that B cell-deficient patients were still infected with human noroviruses, they contained an average log-less virus than did B cell-competent patients (PMID: 26908782). This is entirely consistent with B cells serving as one of multiple cellular targets; and with a 1-log reduction in murine norovirus titers in Rag1-/- mice (PMID: 25378626). I am not aware of any publication, ours or others, that claims B cells are the sole target of noroviruses but there is ample evidence that they can be infected by noroviruses in vitro and in vivo.
- Norovirus frequency on immunocompromised patients presented a seasonality that coincides with that of the community. This suggest that immunocompromised patients are getting the virus circulating in the community and not vice versa as suggested on this and other studies (Karst and Baric Journal of Virology 2015; Vega et al. Journal of Virology 2014). Genetic evidence support the former (Bok et al. Open Forum Infectious Diseases 2016; van Beek et al. Journal of Infectious Diseases 2017). Please review your genetic data and discuss the findings.
Response: Because we observed infections through an extended season beyond what is typically seen in the community, it is possible that both scenarios are supported. In fact, the van Beek et al. study unified these two ideas by showing that while community viruses are acquired by immunocompromises hosts, they evolve to become distinct from circulating variants, which could support the concept of a distinct reservoir of emergent strains. We have now noted these possibilities in the Discussion.
- One major drawback of the study is that it seems (not specified in Materials and Methods sections) that the fold increase of the viral RNA was calculated without using a viral RNA standard for the qRT-PCR. Please provide additional details on how the fold increase was calculated and normalized among the different samples. Notably, approximately 70% of the viruses that authors claim to present viral replication have <10 fold increase. I wonder how authors account for the inherent variability of the assay in stool samples. Please provide evidence on the reproducibility of your detection/quantification method.
Response: As specified in line 150 of the original manuscript, we enumerated viral genome copies and calculated fold-increases as described in Method A of a detailed published protocol in Jones et al. (PMID: 26513671). We always use an RNA standard to generate a standard curve in the qRT-PCR assays, prepared as described on pages 12-13 of the above-mentioned reference. Our data represent the average fold-increase of triplicate reactions per sample tested in a minimum of two independent experiments. Although there was inherent variability between stool samples which has been reported in enteroids as well, there was minimal variability for a given sample as represented by error bars in each graph.
Minor points:
- Line 61: Please correct the statement "Norovirus infections in healthy individuals resolve themselves in a few days [14]." While symptoms can be resolved within days, norovirus shedding has been shown to persist for weeks in healthy individuals (e.g. Atmar et al. Emerging Infectious Diseases 2008; Parra and Green Emerging Infectious Diseases 2014).
Response: We agree with the reviewer that there is a difference between infection and symptoms. The sentence has been edited with the advice of the reviewer.
- Please correct the statement "Until recently, a new GII.4 strain emerged every 2–4 years [9,10]." While that statement is correct for certain strains, the US95 variant emerged in 1995 (Vinje et al. Journal of Infectious Diseases 1997) and was replaced seven years later by the Farmington Hills variant (Lopman et al. Lancet 2004).
Response: We have modified this sentence to indicate that there have been exceptions to this trend.
3. Please comment the reasons that the genotype of only 44% (57/131) of the GII-positive samples analyzed was identified.
Response: Thank you for highlighting an area of unclarity. We have included in the manuscript reasoning for our genotyping numbers.
- Figure 3: Please note that x-axis legend should read "days."
Response: The x-axis has been labeled on figure 3a.
- Did the authors made any attempt to adapt the viruses collected from immunocompromised patients?
Response: We have attempted to passage a subset of the supernatants from HEK293T cultures in an effort to adapt a virus to cell culture growth but have not had luck.
Reviewer 3 Report
This is interesting study showing that the norovirus particles shedding from the immunocompromized patients are infective over long periods of time.
I have two major questions:
1) Is it possible to evaluate the contribution of continuous norovirus infection for the well-being of the patient? For example, is it possible to find a correlation between diarrhea and norovirus-positivity?
2) The authors could better communicate the potential use of vaccines as preventive measure.
Author Response
This is interesting study showing that the norovirus particles shedding from the immunocompromized patients are infective over long periods of time.
Response: We appreciate the reviewer’s interest in our study
I have two major questions:
1) Is it possible to evaluate the contribution of continuous norovirus infection for the well-being of the patient? For example, is it possible to find a correlation between diarrhea and norovirus-positivity?
Response: This is a great question that we hope to address with future prospective studies that will better determine the impact of these infections on the treatment course of our patients. Although gathering this symptom data is possible, albeit subjective since our study is retrospective and would rely heavily on consistent medical chart review/documentation, we feel that the time burden of documenting the symptom status at the timepoints from which all 4644 samples were collected would not enhance the conclusion of our study.
2) The authors could better communicate the potential use of vaccines as preventive measure.
Response: While this would be interesting to consider, vaccines for this virus are still in development and it is unclear to what extent they would be protective in immunocompromised populations.
Reviewer 4 Report
The manuscript by Amy Davis et al., describes the “Infectious norovirus is chronically shed by immunocompromised hosts”. The authors collected stool specimens from immunocompromised pediatric patients with/without acute gastroenteritis symptoms in 2012-2016. Those specimens were screened and identified viral genotypes by RT-PCR. Further, 20 patients with chronic infections determined shedding viral infectivity with cell culture in vitro replication assay methods.
Comments:
Norovirus is a leading cause of acute diarrhea disease of all population in the world. Several factors are currently increasing the challenge posed by norovirus infections to global healthy, one of them is the rapid rate of the genetic and antigenic evolution of circulating noroviruses, which complicates the development of vaccine and treatment.
This manuscript collected serial specimens and determined shedding viral infectivity with cell culture in vitro replication assay methods.to answer the long term shedding of noroviruses in immunocompromised patients, which were consider to be a reservoir and possible origin of transmission, intra-host viral evolution and virus recombination. Viral culture in HEK293T might be the first report success support norovirus replication in vitro.
1. In this study, the authors found 3 patients with double-or triple- sequential infection and 2 co-infection with different genotype viruses depends on viral ORF2 genotypes (line 216-225). These cases cannot find correspond information in Table 3.
2. The authors may analysis viral genome (ORF1 and ORF2) evolution or recombination of sequential specimens between each episode in these cases.
3. In Fig 4, human cell lines, BJAB, HEK293T and SY5Y, all susceptible to human norovirus infection and replication with norovirus genotypes GII.6 and GII.4 infection. Why authors choose HEK293T as the cell line for all samples infectivity testing?
4. According to Fig 5, the viral load of shedding were not progressive decrease post onset in these pediatric chronic infection cases. The data is different from previous manuscripts, such as in long-term care facility nursing home. What kinds of factors may influence the viral shedding or data presentation?
Author Response
The manuscript by Amy Davis et al., describes the “Infectious norovirus is chronically shed byimmunocompromised hosts”. The authors collected stool specimens from immunocompromised pediatric patients with/without acute gastroenteritis symptoms in 2012-2016. Those specimens were screened and identified viral genotypes by RT-PCR. Further, 20 patients with chronic infections determined shedding viral infectivity with cell culture in vitro replication assay methods.
Comments:
Norovirus is a leading cause of acute diarrhea disease of all population in the world. Several factors are currently increasing the challenge posed by norovirus infections to global healthy, one of them is the rapid rate of the genetic and antigenic evolution of circulating noroviruses, which complicates the development of vaccine and treatment.
This manuscript collected serial specimens and determined shedding viral infectivity with cell culture in vitro replication assay methods.to answer the long term shedding of noroviruses in immunocompromised patients, which were consider to be a reservoir and possible origin of transmission, intra-host viral evolution and virus recombination. Viral culture in HEK293T might be the first report success support norovirus replication in vitro.
- In this study, the authors found 3 patients with double-or triple- sequential infection and 2 co-infection with different genotype viruses depends on viral ORF2 genotypes (line 216-225). These cases cannot find correspond information in Table 3.
Response: We apologize for lack of clarity. We have carefully reviewed the data to ensure table 3 coincides with the text.
- The authors may analysis viral genome (ORF1 and ORF2) evolution or recombination of sequential specimens between each episode in these cases.
Response: While this would be interesting to evaluate, studying the evolution/recombination of viruses during the patients’ episodes was not the focus of this study and unfortunately, after giving best efforts to genotype many of the samples, there is no longer fecal material remaining. However, this could certainly be explored by future studies, particularly those with a prospective design.
- In Fig 4, human cell lines, BJAB, HEK293T and SY5Y, all susceptible to human norovirus infection and replication with norovirus genotypes GII.6 and GII.4 infection. Why authors choose HEK293T as the cell line for all samples infectivity testing?
Response: Because we wanted to test as many virus-positive stools as possible, we selected the cell line that grows the most readily and reliably in our hands.
- According to Fig 5, the viral load of shedding were not progressive decrease post onset in these pediatric chronic infection cases. The data is different from previous manuscripts, such as in long-term care facility nursing home. What kinds of factors may influence the viral shedding or data presentation?
Response: While we did not focus on this particular concept, it would appear that virus levels decrease at later time points in comparison to earlier time points in Figure 5. However, we agree that this would not be considered a “progressive decrease” for all patients. We are hesitant to compare our study to these previous studies in long-term care facilities given that we measured infectious virus in cell culture rather than total virus genomes based on the fecal sample (PMID: 18417655, 23153821). For this reason, our readout is much more sensitive to factors that could inhibit viral replication, including chemotherapy drug treatments that would be present in excreta or even issues surrounding fecal collection/storage that we are unable to adjust for retrospectively.
We appreciate the opportunity to revise our manuscript, which allowed us to fix additional typographical errors encountered during the response to revisions.
Round 2
Reviewer 2 Report
Thank you for including my comments in your revised version.
1. The authors "agree that a more detailed follow-up study of human norovirus replication in HEK293T and SY5Y cells is warranted" and infectious virus from chronically-infected immunocompromised patients was shown with the esteroid system. This reviewer do not see the reason why to rush the publication of the HEK293T and SY5Y cells as in vitro system for human noroviruses without providing any evidence on the production of viral proteins or novel virions.
2. While authors provided evidence that infection virus is shed by immunocompromised patients and hence additional measurements might be needed to restrict transmission, the hypothesis that those patients are the source of pandemic norovirus strains lacks: empirical evidence that chronically-infected immunocompromised patients transmit the virus to healthy individuals, and any evidence that emerging variants was first seem in immunocompromised patients. Genetic evidence in fact supports the opposite (Bok et al. Open Forum Infectious Diseases 2016; van Beek et al. Journal of Infectious Diseases 2017). Although there are reports showing that viruses from immunocompromised patients "evolve to become distinct from circulating variants" mathematical models have shown that "such hosts are rare and tend to be isolated" (Eden et al. Virus Evoluion 2017). Moreover, recently Tohma et al mBio 2019 have shown that viruses distinct from those circulating GII.4 variants can be detected in the community and are not linked to any increase in the number of norovirus infections in further years. Please discus those two points in fairness of the opposite hypothesis (i.e. immunocompromised patients not linked to the emergence of new pandemic viruses).
Author Response
Thank you for including my comments in your revised version.
We appreciate the comments and are glad that they are satisfactory.
- The authors "agree that a more detailed follow-up study of human norovirus replication in HEK293T and SY5Y cells is warranted" and infectious virus from chronically-infected immunocompromised patients was shown with the esteroid system. This reviewer do not see the reason why to rush the publication of the HEK293T and SY5Y cells as in vitro system for human noroviruses without providing any evidence on the production of viral proteins or novel virions.
Response: We understand the reviewer’s concern; however, the overall goal of the study was not to validate a new in vitro system. Rather, we sought to address whether chronically shed virus in patient samples remained infectious. Screening these samples in HEK293T cells offered a way to do this in a quick, cost-effective, and consistent manner that we validated with the enteroid system. All in vitro systems for norovirus have limitations and additional studies of the various methods with head-to-head comparisons is warranted, but such studies would not change or add to the findings of this study. These points of discussion have now been included and we have modified the wordage of our manuscript to reflect the preliminary state of these methods.
- While authors provided evidence that infection virus is shed by immunocompromised patients and hence additional measurements might be needed to restrict transmission, the hypothesis that those patients are the source of pandemic norovirus strains lacks: empirical evidence that chronically-infected immunocompromised patients transmit the virus to healthy individuals, and any evidence that emerging variants was first seem in immunocompromised patients. Genetic evidence in fact supports the opposite (Bok et al. Open Forum Infectious Diseases 2016; van Beek et al. Journal of Infectious Diseases 2017). Although there are reports showing that viruses from immunocompromised patients "evolve to become distinct from circulating variants" mathematical models have shown that "such hosts are rare and tend to be isolated" (Eden et al. Virus Evoluion 2017). Moreover, recently Tohma et al mBio 2019 have shown that viruses distinct from those circulating GII.4 variants can be detected in the community and are not linked to any increase in the number of norovirus infections in further years. Please discus those two points in fairness of the opposite hypothesis (i.e. immunocompromised patients not linked to the emergence of new pandemic viruses).
Response: We appreciate the reviewer’s support of this point and have toned down aspects of this discussion in our revised manuscript. With respect to the studies by Eden et al. and Tohma et al., unfortunately these were not designed or focused on immunocompromised hosts within a hospital setting. The mathematical model by Eden et al. notes that they “do not account for detailed host population structures that do exist in an institutional setting like hospitals and nursing homes, and may act as hubs for transmissions between immunocompromised patients and subsequently increase their relative contribution” and therefore “assume equal underlying transmission rates for each of our subpopulations.” Clearly, these are critical factors that could drastically change the transmission dynamics within a patient population such as the one we investigated in our study. In Tohma et al. they note that their analysis was restricted to “a total of 1,601 full-length (1,623-nt) and nearly full-length (1,560 nt) VP1 sequences of the GII.4 genotype, from which sequences from immunocompromised patients and environmental samples were removed, were downloaded from GenBank (accessed July 2017),” so their findings may not be generalizable to immunocompromised populations.